materials science

silicone cross-linked polyethylene, micro-rotation rheological equipment, off-axial direction, hoop torsional strength

**Author for correspondence:**
Shibing Bai
e-mail: baishibing@scu.edu.cn

This article has been edited by the Royal Society of Chemistry, including the commissioning, peer review process and editorial aspects up to the point of acceptance.

# Preparation of high-performance polyethylene tubes under the coexistence of silicone cross-linked polyethylene and rotation extrusion

Fasen Sun[1], Jia Guo[2], Yijun Li[1], Shibing Bai[1] and Qi Wang[1]

[1]State Key Laboratory of Polymer Materials Engineering, Polymer Research Institute of Sichuan University, Chengdu 610065, People's Republic of China
[2]State Key Laboratory of Special Functional Waterproof Materials, Beijing Oriental Yuhong Waterproof Technology Co., Ltd, No. 2 Shaling Section, Shunping Road, Beijing 100020, Peoples' Republic of China

(iD) FS, 0000-0002-3014-9215

In this study, the silicone cross-linked polyethylene (Si-XLPE) powder with better thermoplastic performance and abundant cross-linked network points was attained by using solid-state shear mechanochemical ($S^3M$) technology and it was added into high-density polyethylene (HDPE) matrix to prepare Si-XLPE/HDPE tubes by a rotation extrusion rheometer. SEM and 2D-SAXS experiments showed that the presence of Si-XLPE and rotation extrusion facilitated the formation of stable shish-kebabs which deviated from the axial direction in polyethylene (PE) tubes. This result was interpreted that introduction of Si-XLPE in PE tubes provided abundant molecular cross-linked network structures, which suppressed the thermal movement and relaxation of oriented molecular chains owing to intermolecular interaction. Moreover, the axial and hoop flow field further promoted orientation of the permanent cross-linked network entanglement points and formation of more stable cluster-like shish structures in the off-axial direction during the rotation extrusion process. Besides, our experimental results had also ascertained that molecular orientation and shish-kebabs in off-axial direction should be the primary responsibility for the remarkable enhancement of hoop torsional strength in PE tubes. Hoop torsional strength of PE tubes adding Si-XLPE reached 19.58 MPa when the mandrel rotation rate was 30 r.p.m., while that of conventional extruded PE tubes was only 9.83 MPa. As a consequence, PE tubes with excellent performance were prepared under the combined effect of Si-XLPE and rotation extrusion.

# 1. Introduction

Polymer tubes are used in a variety of medical and industrial procedures, and have attracted more and more attention in clinical applications and mechanical parts due to their outstanding properties, such as low resistance to flow, light weight, high resistance to corrosion and ease of manufacture [1–3]. Polyethylene (PE) tubes are one of the common polymer tubes and offer a variety of uses in the mechanical setting, including the protection and delivery of miniature components (such as electricity and heat conductive, signal and microscopic materials) inside the equipment [4]. So, the introduction of PE tubes greatly reduces the damage to internal components and avoids the troubles caused by operation and assembly–disassembly procedure. For most operation and detection procedures, lower kink-resistance of PE tubes increases the likelihood of migration of tubes from the target location, resulting in adverse effects of entity analysis [5,6]. In addition to the migration of tubes, insertion of PE tubes can be difficult along a tortuous path and needs multiple adjustments to be correctly placed [7]. So the excellent kink-resistance has the vital practical significance for industrial application of PE tubes [8,9].

Incorporation of inorganic reinforcing fillers is an effective way to improve the properties of polymer materials [10–13]. Wang *et al.* reported that the electrospun fibres could act not only as reinforcing fillers but also as superior nucleating agents to alter the crystal morphology of the polypropylene (PP) matrix [14]. Nevertheless, addition of the inorganic filler in polymer melts induced the interface instability of the polymer matrix, and slipping of the interface resulted in poor compatibility of both phases, which finally affected the optimal mechanical properties of the polymer. Compared with that of the inorganic filler, self-enhancement of polymer became a mainstream subject due to the absence of compatibility and interface issues. As we all know, the internal crystalline structures and morphology, which are decided by the thermo-mechanical history and moulding way, play a significant role in the performance of polymer products [15,16]. Therefore, an effective way to reinforce polymeric properties is to regulate crystalline structures and morphology in PE tubes during the moulding processes, such as molecular orientation and shish-kebab structures.

For conventional moulding process, PE melts are axially extruded along the annular cavity, and so molecular orientation is paralleled to the axial direction, leading to low efficiency on hoop torsional properties of PE tubes [17,18]. Because of the axial orientation of molecular chains, polymer tubes display poor properties to hoop load and thus are more likely to twist occlusion, leading to ineffective material transportation [19,20]. So, the axially aligned reinforcement goes against kink-resistance of PE tubes. In this regard, novel rotation extrusion equipment has been designed and developed by our group, in which the mandrel and die can rotate independently by adjusting the corresponding motor and gear reduction system when polymer melts are extruded [21]. During the rotation extrusion process, with superposition of the axial flow and hoop drag flow caused by the mandrel and die rotation, polymer melts flow by a helical mode to induce the formation of molecular orientation and shish-kebabs deviating from the axial direction in the annulus passage [22–24]. Recently, the new rotation rheometer has been designed on the basis of previous ideas of rotation extrusion equipment and could prepare micro-functional tubes. For example, Zhang *et al.* [25] reported that $PP/Ag/TiO_2$ composite tubes prepared by mandrel rotation exhibited the desired properties with the enhanced kink-resistance and superior antibacterial activity. Li *et al.* [26] also studied that the multiaxial orientation generated by the combined effect of carbon fibres and rotation extrusion obviously promoted electrical conductivity and sensitive strain-responsive performance in low-density polyethylene and carbon fibre (LDPE/CF) tubes. Therefore, it is an extraordinary moulding way to form molecular orientation and shish-kebabs deviating from the axial direction in polymer tubes [27,28].

During the polymer moulding process, the polymer melts are extruded under complex and dense flow fields which, in turn, affect the subsequent crystallization and the final crystalline morphology and properties of the materials [29,30]. Micro-extrusion tubes are susceptible to internal crystalline structures and morphology generated in the moulding process [31]. However, the oriented molecular chains and off-axial orientation inside polymer tubes easily reach a coiled state and there was no or only a small amount of shish-kebabs in off-axial direction for the final polymer products due to thermal motion and relaxation of molecular chains during the rotation extrusion process [32]. Accordingly, to manipulate the orientation alignment and crystalline structures in micro-PE tubes is an essential condition for securing its application. It is extremely important and challenging to prevent the molecular chains from relaxation and increase the oriented molecular chains deviating from the axial direction in PE tubes.

As we all know, long molecular chains and entangled network structures could slow down relaxation of molecular chains and facilitate the formation of shish-kebabs. Long polymer molecular chains or

entangled network structures, which are more easily extended and undergo coil-stretch transition than short molecular chains under external flow fields because of the longer relaxation time, could facilitate stability of the flow-induced orientation state [33]. However, silicone cross-linked polyethylene (Si-XLPE), which is different from peroxide cross-linked polyethylene (P-XLPE) and irradiation cross-linked polyethylene (I-XLPE) materials, cannot be added into low molecular weight PE as the stable cross-linked network component during the external flow field. Owing to the extraordinary three-dimensional cross-linked network structures and unique cross-linking way, Si-XLPE exhibited extra high cross-linking degree, making it harsh to melt processing. For the above reason, our current work was that the Si-XLPE powder obtained by the solid-state shear mechanochemical (S³M) technology was melt blended uniformly with high-density polyethylene (HDPE) and then extruded into the pellets. Compared with the cross-linked polyethylene attained by the other cross-linking methods, the Si-XLPE powder prepared by the S³M technology exhibited excellent processing rheological properties by regulating the content of permanent cross-linked network points. Eventually, the prepared pellets were extrusion-moulded into tubes with excellent kink-resistance to manipulate orientation alignment and shish-kebabs in off-axial direction by the rotation rheometer.

# 2. Experimental sections

## 2.1. Material

The HDPE was the commercially available resin from MaoMing petrochemical Co., Ltd (Guangdong, China). The weight-average molecular weight ($M_w$) was $1.86 \times 10^5 \, \text{g mol}^{-1}$ and the melt flow rate was $0.28 \, \text{g} \, 10 \, \text{min}^{-1}$, measured at $190°C$ under $2.16$ kg.

The Si-XLPE was provided by the Si-XLPE research laboratory, TBEA Deyang Cable Stock Co., Ltd; the gel content was about 65.8% and the melt flow rate was $0.017 \, \text{g} \, 10 \, \text{min}^{-1}$, measured at $210°C$ under $2.16$ kg.

## 2.2. Sample preparation

The Si-XLPE was milled and collected by the S³M technology in different cycles. The operating process was mentioned in the previous literature [34]. The Si-XLPE materials were fed into the centre of the milling pan of the S³M reactor from the inlet at a fixed rotating speed and driven by the shear force, moving along a spiral route toward the edge of the pan until they came out from the discharge port; this completed one cycle of milling. Samples from the discharge port were collected and made ready for the next milling cycle. The repetition operation continued for 10 cycles to prepare the Si-XLPE powder. The particle size and cross-linking degree were controlled by the velocity of the moving pan and imposed load during milling [34]. Meanwhile, particle size, gel content, melt flow index and tensile strength of Si-XLPE with different milling cycles are shown in figure 1. Compared with the contribution of Si-XLPE particle size of 2 and 10 milling cycles, the reduction of particle size could contribute more to uniform dispersion of the Si-XLPE into the PE matrix. Although permanent network structures can suppress relaxation of molecular orientation owing to the strong limited effect endowed by cross-linked network points, the presence of XLPE with high cross-linking degree is unfavourable to the melting processing of polymer. Because cross-linking degree of XLPE is inversely proportional to the flow ability and XLPE with high cross-linking degree exhibited the poor flow ability. The flow ability of Si-XLPE material was reflected by melt flow index (MFI), so the opposite trend is shown in figure 1c. As we all know, cross-linking degree of XLPE is usually evaluated by the gel content. Gel content of Si-XLPE obviously declined with milling cycles increasing as shown in figure 1b. So, Si-XLPE powder with 10 milling cycles not only provided enough cross-linked network points but also exhibited the better flow ability. Compared with Si-XLPE with the different milling cycles, the tensile strength of the 10 cycle samples was better than that of the 0 cycles in figure 1d, implying that the Si-XLPE powder with 10 milling cycles obtained more excellent thermoplastic performance and flow ability. Obviously, Si-XLPE featuring better flow ability easily dispersed evenly in the PE matrix, which reduced aggregation of cross-linked molecular chains during the processing, leading to better performance of the prepared tubes.

2 wt% Si-XLPE with 10 milling cycles was mixed into the HDPE matrix by a twin-screw extruder, and the prepared pellets in molten state were fed into the annular cavity of the rotating head through micro-extrusion equipment. Under the extrusion pressure and tractive effort, hollow tubes were cool

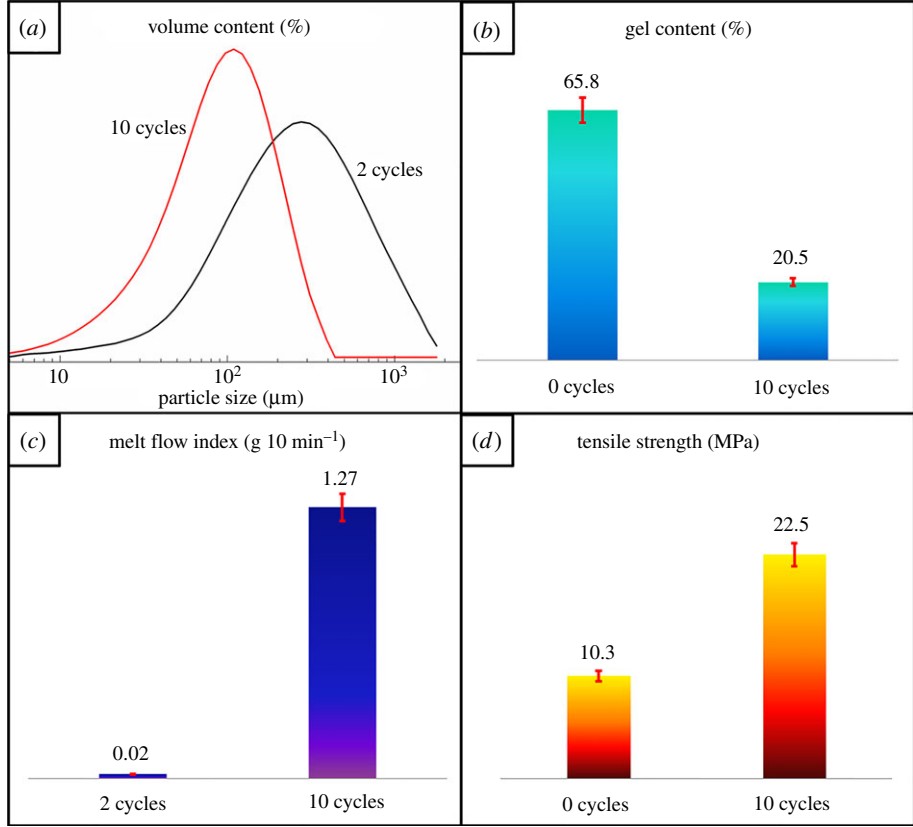

**Figure 1.** Particle sizes, gel content, melt flow index and tensile strength of Si-XLPE with different milling cycles.

formed in an air atmosphere. The detailed description of the rotation rheometer was reported in the supporting information [26]. The tubes with different crystalline morphology and performance were prepared by adjusting the mandrel rotation rates. The prepared tubes were referred to as XPE-Y, and Y represented the mandrel rotation rate. For example, XPE-0 is the rotation extrusion tube adding Si-XLPE under the rotation rate of 0 r.p.m., namely the conventional extrusion tubes. For comparison, rotation-extruded HDPE tubes without Si-XLPE were also prepared at the same processing conditions, designated as PE-Y.

## 2.3. Characterization

For gel content experiment, Si-XLPE powder with different milling cycles was extracted with o-xylene at 140°C for 24 h. The rest of the extraction was taken out from Soxhlet extraction and washed with ethanol repeatedly. Finally, the remaining part was dried at 80°C under vacuum oven for 10 h, so that the solvent would vaporize completely. The percentage ratio of the final weight of the polymer to its initial weight was regarded as the gel content.

Particle sizes of Si-XLPE powder with the different milling cycles (2 and 10 cycles) were analysed by the laser particle size analyser (Mastersizer 2000). The specimens were dispersed by ultrasonic treatment before measurement; ethanol was needed as a dispersion medium during measurement.

According to ASTM-D 1238, the melt flow index experiment of Si-XLPE with different milling cycles was carried out by XNR-400 (Chengde Jinjian Testing Instrument Co. Ltd). The mass of polymer was recorded at a temperature of 210°C under 10 kg loads and in 10 min.

The mechanical properties of Si-XLPE samples with different milling cycles were tested at room temperature in accordance with ASTM D 412 through the universal tensile testing machine (RGM-4010, Shenzhen Reger Instrument Co., Ltd) at a crosshead speed of 50 mm min$^{-1}$. The data were averaged over five specimens within tolerance range.

The melt behaviours of PE tubes were characterized by a differential scanning calorimeter (DSC; Q20, TA Instruments, USA) under nitrogen with a flow rate of 20 ml min$^{-1}$. The specimens were cut from the prepared tubes and accurately weighed about 6–9 mg. The temperature cycle was from 40°C to 180°C with a heating rate of 10°C min$^{-1}$. Then, they were cooled down to 40°C at a constant rate of

30°C min$^{-1}$. The melting curves were recorded and analysed after the experiment. The melting temperatures were defined as the peak temperature, and the crystallinity and lamellar thickness can be obtained by the following equations [35]:

$$X_c = \frac{\Delta H_f}{\Delta H_0} \times 100\% \tag{2.1}$$

and

$$L_c = \frac{2\sigma_e T_m^o}{\Delta h_f (T_m^o - T_m)}, \tag{2.2}$$

where $X_c$ is the crystallinity of samples; $\Delta H_f$ is the melting enthalpy and $\Delta H_0$ is 293 J g$^{-1}$; $L_c$ is the lamellar thickness; $\sigma_e$ is the surface free energy of lamellae and is $9 \times 10^{-6}$ J cm$^{-2}$; $\Delta h_f$ is the melting enthalpy per unit volume and is 293 J cm$^{-3}$; $T_m^0$ is the equilibrium melting temperature of 418 K; $T_m$ is the melting temperature.

X-ray diffraction measurements of the samples were taken by an X'Pert X-diffractometer (XRD; Philips Analytical BV, Almelo, The Netherlands) with Cu K$\alpha$ radiation at k$\frac{1}{4}$ 0.1540 nm (40 kV, 40 mA). The crystallinity degree ($X_{c,R}$) was obtained by the resolution of XRD patterns into the diffraction area relative to the crystalline peaks ($I_c$) and amorphous halo ($I_a$),

$$X_{c,R} = \frac{I_c}{I_c + I_a} \times 100\%. \tag{2.3}$$

The crystalline morphology of the prepared tubes was investigated by an Inspect F50 SEM instrument (FEI Co., The Netherlands) at 20 kV. Prior to the observation, samples were cut along the axial direction and etched chemically according to the potassium permanganate etching technique described in the literature [32] to remove the amorphous phase. Finally, the surface of the etched samples was coated with a thin layer of gold for observation by the SEM.

The samples were cut along the axial direction and monitored by *in situ* two-dimensional (2D) small-angle X-ray scattering (SAXS) measurement. The X-ray wavelength was 0.154 nm and a Mar165 CCD detector was employed to collect time-resolved 2D-SAXS patterns. The exposure time was 5 min and sample-to-detector distance was calibrated to be 2481 mm. Meanwhile, the lamellar thickness ($L_c$) was calculated by the long period ($q_{max}$) from SAXS patterns,

$$L_c = \frac{2\pi}{q_{max}}. \tag{2.4}$$

Melt rheological experiments were conducted by an AR2000EX rotational rheometer (TA Instruments, USA) having the parallel plates with a diameter of 25 mm and a gap of 2 mm at constant temperatures in a nitrogen atmosphere. The sample was heated at a constant temperature and kept for 5 min. Then dynamic frequency sweep measurements were performed by an angular frequency range at 190 and 210°C.

Thermal shrinkage was carried out to estimate the molecular orientation in PE tubes. Along the axial direction, the samples of 15 mm length were cut from the prepared tubes, then put into glycerol and kept at 145°C until no dimensional changes occurred. The lengths of tubes before and after heating were measured and the shrinkage ratio was calculated according to the ratio of original-to-final dimension.

The hoop torque test is used to characterize the hoop torsional properties of tubes. Torsional strength of the prepared tubes was measured in a torque dynamometer (MTT03-50Z, America MARK-10 Co. Ltd). The 40 mm length samples were cut from the flow direction of the prepared tubes and clamped in a test table and tested under 1.7 rad min$^{-1}$ of rotational speed. The standard length of samples was 20 mm in the test. The average value of five specimens was recorded.

# 3. Results and discussion

## 3.1. Effect of Si-XLPE on crystalline modification during rotation extrusion

The performance of PE tubes was influenced inevitably by crystalline structures, which can be obtained by both DSC and XRD, as shown in figure 2. Apparently, the presence of a small amount of Si-XLPE hardly affected the characteristic melting curves and XRD patterns of the samples. More information on lamellar thickness ($L_c$) and crystallinity ($X_c$) was calculated by the DSC curves and summarized in

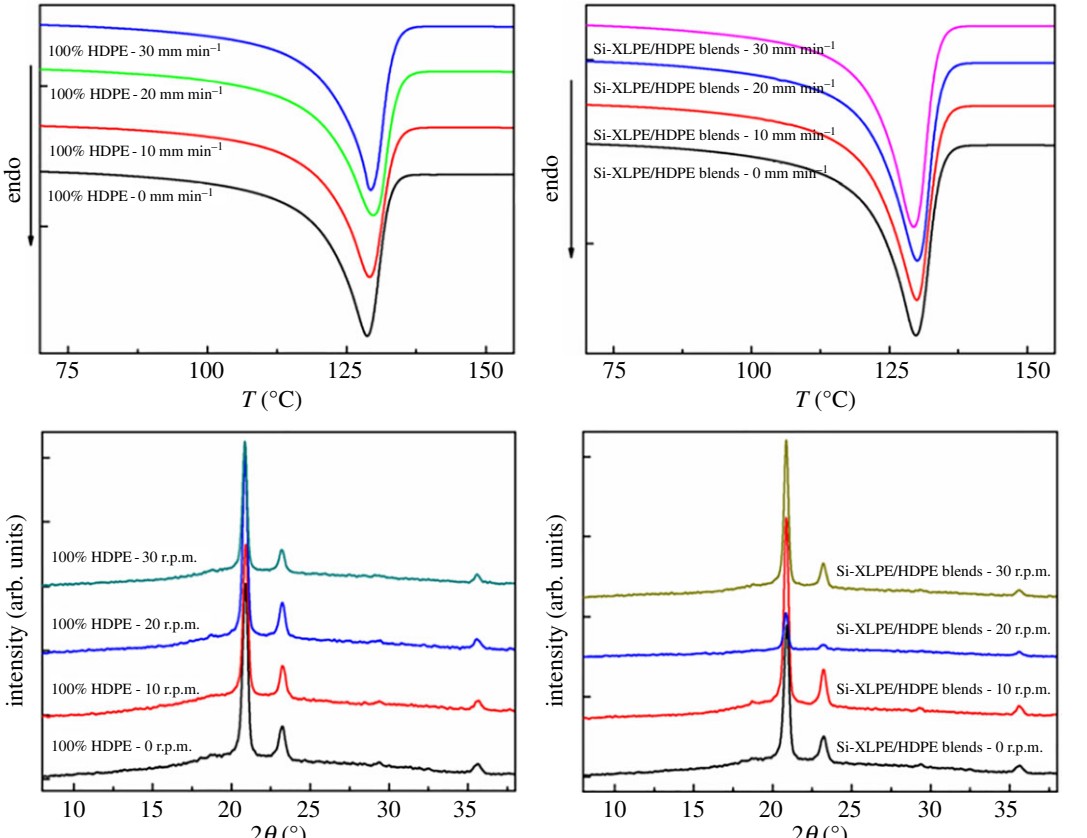

**Figure 2.** DSC curves and XRD patterns of PE and XPE tubes as a function of mandrel rotation rates.

table 1. Moreover, there was only a slight change in lamellar thickness and crystallinity with the increase of mandrel rotation rates, demonstrating that rotation extrusion did not change the crystalline modification of the PE matrix [36,37]. The crystallinity ($X_c$) and lamellar thickness ($L_c$) of PE and XPE tubes were obtained by XRD and SAXS, respectively, as shown in table 2. It is interesting that although the crystalline data obtained from DSC and XRD slightly differed from each other, both data displayed the same trend that with increasing rotation rate, the crystallinity and lamellar thickness firstly increased and then slightly dropped once the rotation rate exceeded 30 r.p.m. Since the crystallization of polymers involves the ordered alignment of the molecular chains, higher stress can facilitate the ordering procedure and thus increase crystallinity and lamellar thickness. Nevertheless, high-speed rotation could generate a large amount of shear heat to facilitate the stretched molecular chains to relax back into the random state. So, both experimental results indicated that Si-XLPE and mandrel rotation hardly showed the difference of crystalline structures in PE and XPE tubes.

## 3.2. Effect of Si-XLPE on orientation during rotation extrusion

Thermal shrinkage could indirectly characterize the variation of flow-induced orientation structures in the condensed state of polymer materials. When the samples were heated above their melting temperature, the oriented molecular chains would relax back to the random coiled state and thus the length along the orientation direction would shorten. Therefore, the shrinkage ratio could be considered as an indicator of orientation structures for polymer materials [38]. Figure 3 shows photographs of strips with original 15 mm length after heating and thermal shrinkage ratio for PE and XPE tubes with the different mandrel rotation rates. For conventional extruded tubes in figure 3a,c, the shrinkage ratio of PE and XPE tubes was 5.8 and 4.1, respectively. Clearly, the higher axial orientation existed in conventional extruded tubes, because polymer melts suffered the axial squeezing force and flowed along the longitudinal direction so that molecular chains extended and aligned along the axial direction. Besides, the axial shrinkage of rotation-extruded tubes was less than that of conventional extruded tubes as shown in figure 3b,d, and the shrinkage ratio of rotation-extruded tubes obviously decreased with the increase of mandrel rotation rates. This result

**Table 1.** $T_m$, $\Delta H$, crystallinity and lamellar thickness of PE and XPE tubes with different mandrel rotation rates by DSC curves. $X_{c,D}$: crystallinity obtained based on DSC curves; $L_c$: lamellar thickness calculated DSC curves.

| samples | $T_m$ (°C) | $\Delta H$ (J/g) | $X_{c,D}$ | $L_c$ (nm) |
|---|---|---|---|---|
| PE-0 | 128.67 ± 0.24 | 151.4 ± 0.3 | 0.517 ± 0.001 | 15.73 ± 0.23 |
| PE-10 | 129.12 ± 0.31 | 151.6 ± 0.4 | 0.518 ± 0.002 | 16.18 ± 0.31 |
| PE-20 | 129.77 ± 0.26 | 156.2 ± 0.2 | 0.533 ± 0.001 | 16.86 ± 0.28 |
| PE-30 | 129.32 ± 0.19 | 155.9 ± 0.4 | 0.532 ± 0.002 | 16.38 ± 0.2 |
| XPE-0 | 129.82 ± 0.32 | 154.1 ± 0.2 | 0.526 ± 0.001 | 16.92 ± 0.36 |
| XPE-10 | 129.97 ± 0.14 | 154.7 ± 0.4 | 0.528 ± 0.002 | 17.09 ± 0.16 |
| XPE-20 | 130.03 ± 0.43 | 159.1 ± 0.2 | 0.543 ± 0.001 | 17.17 ± 0.49 |
| XPE-30 | 129.42 ± 0.2 | 156.7 ± 0.3 | 0.535 ± 0.001 | 16.49 ± 0.21 |

**Table 2.** $q_{max}$, crystallinity and lamellar thickness of PE and XPE tubes with different mandrel rotation rates by XRD and SAXS curves. $X_{c,R}$: crystallinity obtained by XRD patterns; $L_c$: lamellar thickness calculated SAXS results.

| samples | $q_{max}$ | $X_{c,R}$ | $L_c$ (nm) |
|---|---|---|---|
| PE-0 | 0.322 ± 0.005 | 0.56 ± 0.003 | 19.51 ± 0.3 |
| PE-10 | 0.319 ± 0.006 | 0.58 ± 0.002 | 19.69 ± 0.37 |
| PE-20 | 0.302 ± 0.01 | 0.597 ± 0.002 | 20.82 ± 0.69 |
| PE-30 | 0.303 ± 0.006 | 0.591 ± 0.001 | 20.74 ± 0.41 |
| XPE-0 | 0.319 ± 0.004 | 0.602 ± 0.002 | 19.69 ± 0.25 |
| XPE-10 | 0.313 ± 0.007 | 0.611 ± 0.001 | 20.07 ± 0.45 |
| XPE-20 | 0.299 ± 0.004 | 0.617 ± 0.001 | 21.01 ± 0.28 |
| XPE-30 | 0.302 ± 0.003 | 0.607 ± 0.002 | 20.8 ± 0.2 |

was interpreted that rotation extrusion restricted effectively axial orientation of molecular chains, which was attributed to the complicated flow pattern in the rotation extrusion processes. During micro-rotation extrusion, polymer melts suffered from the hoop shear force provided by the mandrel rotation superimposed on the axial squeezing force, which led to a helical flow of polymer melts deviating from the longitudinal direction. Therefore, the special helical flow induced by rotation extrusion triggered the transformation from axial to off-axial direction for orientation structures inside tubes, displaying that the shrinkage ratio of rotation-extruded tubes decreased. Compared with PE tubes, XPE tubes exhibited less shrinkage ratio at the same mandrel rotation rates. When the mandrel rotation rate reached 20 r.p.m., the shrinkage ratio of the PE and XPE tubes was 2.23 and 1.6, respectively. Introduction of Si-XLPE provided abundant permanent cross-linked network structures, which further hindered the molecular chains to relax into the axial direction and formed the orientation structures in the off-axial direction.

Thermal shrinkage analysis was the macroscopic evidence for characterizing the orientation structures, which demonstrated that the combined effects of the mandrel rotation and Si-XLPE promoted the formation of orientation structures in the off-axial direction. SEM measurements were taken to further investigate the crystalline orientation, which is shown in figure 4. For both PE tubes, the anisotropic structures consisting of some cluster-like crystallites and numerous folded molecular chain lamellae that were stacked in parallel and grew perpendicular to these cluster crystallites were found, in accord with the features of typical shish-kebab morphology described by previous studies [39,40]. Essential evidence that revealed orientation of the submicroscopic structures and evolution of crystalline morphology caused by the different rotation extrusion rates came from 2D-SAXS patterns. Figure 4 shows the initial SAXS patterns of PE and XPE tubes with different mandrel rotation rates. As shown in figure 4, the bright spots were found around the equator of all patterns, which was the important evidence that oriented lamellae were exhibited along the meridian direction in polymer materials.

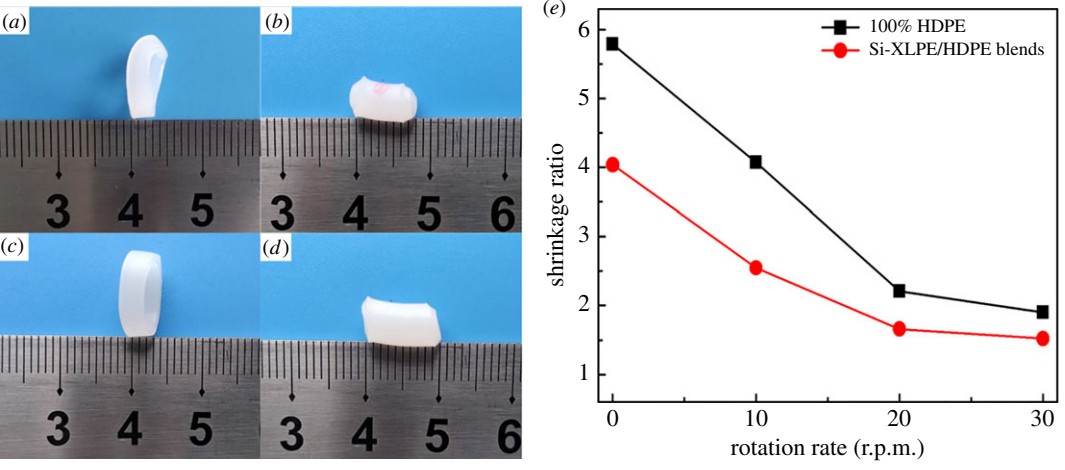

**Figure 3.** Thermal shrinkage photographs (*a*: PE-0, *b*: PE-30, *c*: XPE-0, *d*: XPE-30) and shrinkage ratio (*e*) of PE and XPE tubes with different mandrel rotation rates.

Meanwhile, orientation parameters of lamellae in PE tubes were calculated based on the Herman equation [41] and listed in table 3,

$$f_c = \frac{3\langle\cos^2\varphi\rangle - 1}{2} \tag{3.1}$$

and

$$\langle\cos^2\varphi\rangle = \frac{\int_0^{\pi/2} I(\varphi)cos^2\varphi\sin\varphi \ \mathrm{d}\varphi}{\int_0^{\pi/2} I(\varphi)\sin\varphi \ \mathrm{d}\varphi}, \tag{3.2}$$

where $\varphi$ is the azimuthal angle, $I(\varphi)$ is the intensity of the azimuthal angle, the description about $f_c$ is as follows: $f_c$ presents orientation parameters with the range from $-0.5$ to $1.0$. When $f_c = 1$, the scattering focuses on the perpendicular direction of the flow direction and the orientation is parallel to the flow direction; when $f_c = -0.5$, the scattering focuses on the parallel direction and the orientation is perpendicular to the flow direction; when $f_c = 0$, it means that the sample is perfect isotropic.

Compared with PE tubes, XPE tubes exhibited higher crystalline orientation degree from 0.41 to 0.48 when the mandrel rotation rate was 30 r.p.m. The higher orientation degree for XPE tubes implied that more stable shish-kebabs existed in tubes, which was in accordance with SEM results. Obviously, this result was due to the fact that the presence of chemical cross-linked network structures provided by Si-XLPE could facilitate the formation of more stable shish structures than that of molecular chains in PE tubes during the micro-rotation extrusion process. The shish-kebabs therein comprised the stretched cross-linked network molecular chain beams provided by Si-XLPE as shish and the oriented chain-folded lamellae which grew kebabs along the flow direction. Therefore, the formation of more stable shish-kebabs in XPE tubes could promote the crystalline orientation and improve the orientation degree. As shown in table 3, the hoop shear force induced by the mandrel rotation could promote the formation of shish-kebabs and increase the orientation degree with the increase of mandrel rotation rates. However, the higher mandrel rotation rate restrained the growth of oriented molecular chains and shish-kebabs in PE tubes. While increasing the rotation rate, a large amount of heat was generated inside the PE due to the shear between mandrel and melt. Since polymer is a poor conductor of heat, shear heat was keen to concentrate in the local area, and thus accelerated the thermal motion of molecular chains where the stretching molecular chains relaxed into the random coil [36,37]. This discussion could also explain the variation trend from the DSC and XRD curves.

As shown in figure 4, it was significantly observed that there were variations of the off-axial angle of oriented lamellae in PE and XPE tubes with the different mandrel rotation rates. For PE-0 and XPE-0 tubes in figure $4a_1, a_2$, the alignment of oriented lamellae was parallel to the axial direction of tubes. However, the off-axial alignment of oriented lamellae was observed in other tubes in figure 4 and the off-axial angles increased with the increase of mandrel rotation rates, which could be attributed to flow manipulation of rotation extrusion. By combining the axial flow (caused by extrusion) and the hoop shear flow (caused by mandrel rotation), the squeezing and shearing force that suffered polymer melts triggered the evolution of the flow pattern from the axial flow to the helical flow in the hollow

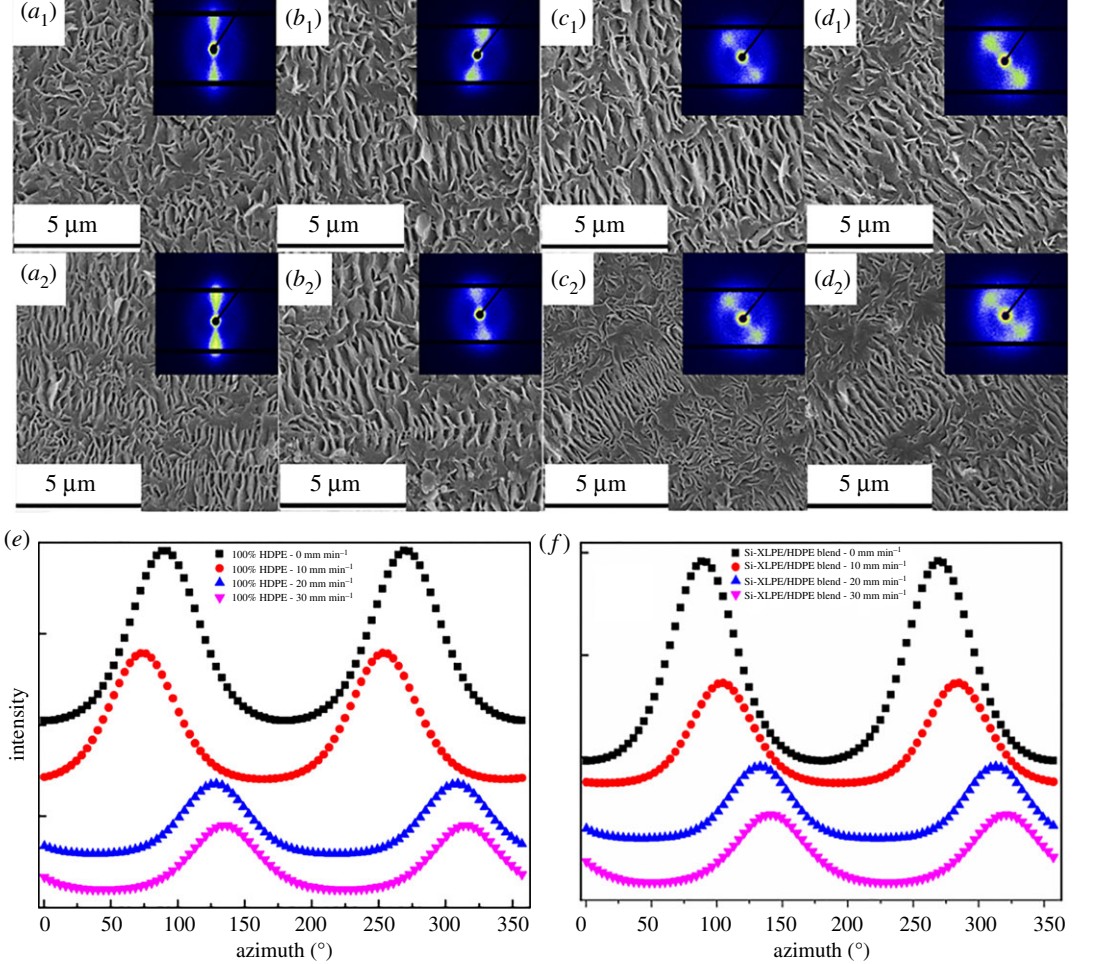

**Figure 4.** The internal crystalline morphologies and 2D-SAXS patterns of PE ($a_1,b_1,c_1,d_1$) and XPE ($a_2,b_2,c_2,d_2$) tubes, and fitted azimuth degree of PE (*e*) and XPE (*f*) tubes with different mandrel rotation rates.

**Table 3.** Orientation parameters of PE and XPE tubes with different mandrel rotation rates.

| samples | 0 r.p.m. | 10 r.p.m. | 20 r.p.m. | 30 r.p.m. |
|---|---|---|---|---|
| PE tubes | 0.32 | 0.42 | 0.46 | 0.41 |
| XPE tubes | 0.4 | 0.46 | 0.5 | 0.48 |

passage during the rotation extrusion process. The anisotropic structures deviated from the axial direction were found in PE tubes.

The complicated axial flow and hoop flow field could promote the alignment and crystallization of the entangled molecular chains and form thermodynamically stable shish-kebab structures in the off-axial direction. Figure 4*e,f* shows the intensity distribution of fitted azimuth degree of PE and XPE tubes as a function of mandrel rotation rate and the off-axial angle of oriented lamellae, as listed in table 4. With the increase of mandrel rotation rates, the off-axial angle of oriented lamellae in PE and XPE tubes obviously increased. Apparently, the alignment of oriented lamellae was more likely to deviate from the axial direction owing to the greater hoop shear force when mandrel rotation enhanced. Interestingly, there was a greater off-axial angle of oriented lamellae for XPE tubes than that for PE tubes at the same rotation rate. For example, the axial-deviating angle of oriented lamellae for XPE-30 tubes was higher than that of PE-30 tubes, as shown in figure 4$d_1,d_2$. Addition of chemical cross-linked network structures in XPE tubes, which could maximize the off-axis effect caused by the combined flow field, should be responsible for the difference in off-axial angle. So, XPE tubes had a greater off-axial angle than that of PE tubes with the help of Si-XLPE and rotation extrusion.

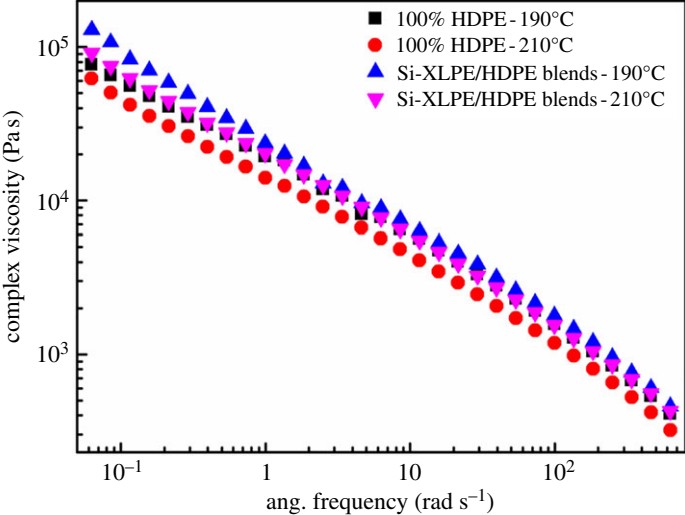

**Figure 5.** Melt rheological behaviours of HDPE and Si-XLPE/HDPE blends at different temperatures.

**Table 4.** Deviation angle (°) of PE and XPE tubes with different mandrel rotation rates.

| rotation rate | 0 r.p.m. | 10 r.p.m. | 20 r.p.m. | 30 r.p.m. |
|---|---|---|---|---|
| PE tubes | 0.5 | 16 | 38 | 43 |
| XPE tubes | 0.5 | 13 | 43 | 51 |

As well demonstrated, as the crucial factors the cross-linked network structures in XPE tubes strongly affected the formation of stable shish-kebab structures and off-axial effect. The formation of shish involves two competing procedures, i.e. the stretch and relaxation of the molecular chains. So, the inherent relaxation time of molecular chains in polymer materials could be used to evaluate the formation of shish-kebab structures in a given flow field [42]. Rheological study of polymer melts, as a common and effective method, analysed the relaxation behaviour of molecular chains [43]. Figure 5 shows the complex viscosity of pure PE and Si-XLPE/PE blend as a function of angle frequency of 190 and 210°C. The complex viscosity of Si-XLPE/PE blend was considerably higher than that of pure PE, as shown in figure 5. In other words, the presence of Si-XLPE contributed to the increment of viscosity for Si-XLPE/PE blend. The cross-linked network points provided by Si-XLPE could promote the molecular chain interactions and further restrain the movement of molecular chains in polymer melts so that the viscosity of polymer melts enhanced [32].

Besides, the mutual competition of orientation and relaxation of molecular chains strongly influenced the lifetime of shish in shish-kebab structures. So, the long relaxation time of molecular chains was a decisive factor to prevent shish from collapsing during the cooling and solidifying process. The relaxation times of pure PE and Si-XLPE/PE blend were calculated based on the simplified Carreau–Yasuda equation, assuming that $\eta\infty$ is zero [44].

$$\eta = \eta_0[1 + (\lambda\omega)^a]^{n-1/a}, \tag{3.3}$$

where $\eta$ is the complex viscosity; $\eta_0$ is the zero-shear viscosity; $\omega$ presents the shearing frequency in radians per second; $\lambda$ means the relaxation time; $a$ is the 'Yasuda constant' representing the transition from Newtonian to power-law behaviour; $n = 2/11$ [45].

The relaxation time of pure PE and Si-XLPE/PE blend was 0.79 and 2.18 s at 190°C and 0.47 and 1.04 s at 210°C, respectively. The results showed that Si-XLPE/PE blend exhibited slower relaxation behaviour than pure PE. For Si-XLPE/PE blend, the presence of stable chemical cross-linked network structures could prevent molecular thermal motion and suppress relaxation of the oriented molecular chains effectively. This implied that the Si-XLPE/PE blends had enough relaxation time to promote the formation of shish-kebabs. The axial flow and hoop shear flow fields could promote chemical cross-linked network structures aligning and forming more stable cluster-like shish structures during the rotation extrusion process and facilitate the folded molecular chains to crystallize and form kebab

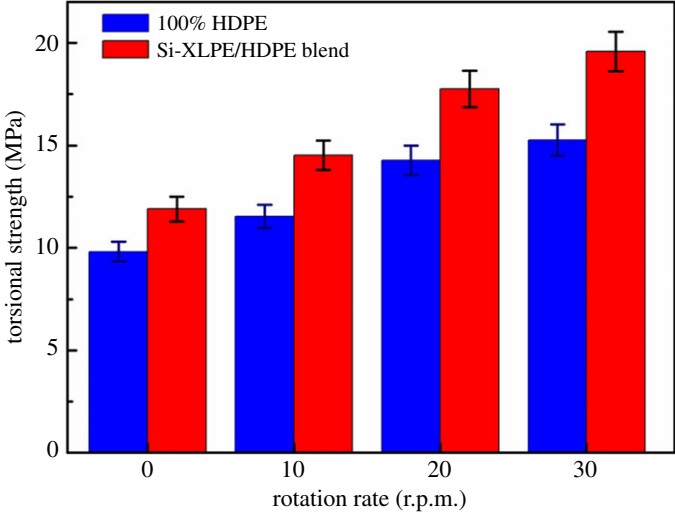

**Figure 6.** Hoop torsional strength of PE and XPE pipes with different mandrel rotation rates.

structures perpendicular to shish, which brought more stable shish-kebab structures. So shish-kebabs induced by chemical cross-linked network points in XPE tubes were stronger and more stable than those induced by molecular chains in PE tubes.

## 3.3. The helix-reinforced behaviour of PE and XPE tubes

Since different crystalline morphology was found in PE and XPE tubes with the different mandrel rotation rates, we were curious whether such superstructure led to the difference of properties in tubes. Hoop torsional strength of PE and XPE tubes with different mandrel rotation rates is presented in figure 6. Clearly, the tubes prepared via rotation extrusion had higher hoop torsional strength than the conventional extruded tubes. Hoop torsional strength of PE and XPE tubes was 9.83 and 11.89 MPa during the conventional extrusion process, respectively. However, hoop torsional strength significantly increased to 15.25 and 19.58 MPa when the mandrel rotation rate reached 30 r.p.m. Compared with that of conventional extruded PE and XPE tubes, hoop torsional strength of rotation-extruded tubes enhanced by 55.1% and 64.7%, respectively. Besides, hoop torsional strength of PE tubes was enhanced with the increase of mandrel rotation rates. This result was the fact that molecular orientation and shish-kebabs in off-axial direction were more easily obtained under the greater shearing force induced by mandrel rotation. In addition, introduction of Si-XLPE could further promote hoop torsional strength of tubes. Because the chemical cross-linked network structures facilitated the formation of more stable shish-kebabs in off-axial direction and maximized the off-axis effect of oriented lamellae. When the mandrel rotation rate was 30 r.p.m., hoop torsional strength of PE tubes increased from 15.25 to 19.58 MPa. Thanks to the combined effects of rotation extrusion and stable cross-linked network structures, the PE tubes showed most excellent performance.

On the basis of the above discussion on the evolution of crystalline morphology and enhancement of torsional strength of PE tubes, we proposed a schematic to elucidate the helix-reinforced mechanism of molecular orientation and shish-kebab structures on the performance for PE and XPE tubes, as shown in figure 7. Although similar crystalline structures were verified by SEM and 2D-SAXS, orientation degree and torsional strength of XPE tubes were better than those of PE tubes. For example, hoop torsional strength of PE-0 tubes increased from 9.83 to 11.89 MPa when Si-XLPE was added into PE matrix during the conventional extrusion process. The presence of permanent cross-linked network structures strengthened the molecular interaction and suppressed the movement and relaxation of oriented molecular chains, which further promote oriented molecular chains forming more stable shish structures, as shown in figure 7. Compared with that of the shish-kebabs induced by molecular chains in PE tubes, the formation of more stable shish-kebabs induced by chemical cross-linked network points of Si-XLPE contributed to the enhancement of torsional strength in XPE tubes. However, hoop torsional strength of XPE-0 tubes only increased by 21% compared with that of PE-0 tubes, so there was less effect of stable shish-kebabs for significant enhancement of mechanical properties in PE tubes. Elimination of axial orientation together with off-axial aligned shish-kebabs should be primarily responsible for the significant enhancement of hoop torsional strength in the rotation-extruded tubes.

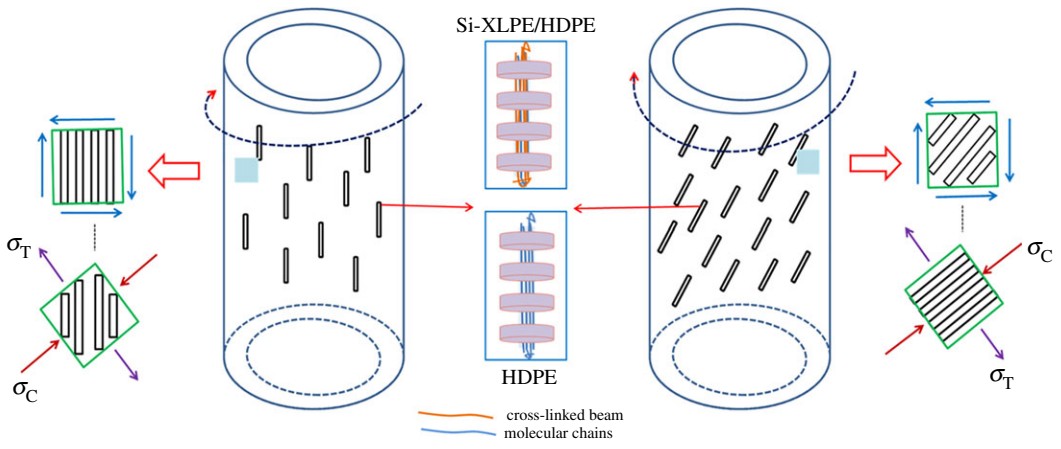

**Figure 7.** Schematic of the helix-reinforced mechanism for PE and XPE tubes with different mandrel rotation rates.

For conventional extruded tubes, the presence of higher axial orientation could not adapt to the off-axial stress environment and bear torque force in the torque testing, as shown in figure 6. When oriented molecular chains aligned along the flow direction and developed into shish-kebabs in off-axial direction during the rotation extrusion process, hoop torsional strength was obviously enhanced. Molecular orientation and shish-kebabs in the off-axial direction could not only withstand the greater hoop stress in the torque process, but also significantly improve hoop torsional strength, and displayed excellent kink-resistance of tubes.

Porter *et al.* [46] had already interpreted that torque testing induced a state of pure shear stress at the surface of helical reinforcement tubes, assuming the stress elements shown could be rotated so that the maximum shear stresses suffered by the helical reinforcement tubes in torque testing became maximum compressive and tensile stresses ($\sigma_C$ and $\sigma_T$, respectively), the enhancement effect of mechanical properties was mainly determined by the compression modulus, as shown in figure 7. When the direction of helical reinforcement phase in tubes was parallel to that of compressive stress, tubes could attain greater modulus from compression modulus by the Voigt model [47] and the mechanical properties were enhanced; conversely, attaining lower modulus from compression modulus by the Reuss model [47], the mechanical properties declined. Accordingly, when the torsional stress was applied to rotation-extruded PE tubes, shish-kebabs in off-axial direction exhibited higher orientation in the direction of compression stress and the compression modulus was enhanced, which improved kink-resistance of tubes. For conventional extrusion tubes, alignment of shish-kebabs was absent in the direction of compression stress, which not only reduced the enhancement effect of shish-kebabs in the compression direction but also triggered the tie-molecular chains attached to shish-kebabs to bear greater tensile stress perpendicular to the flow direction in figure 7, resulting in poor kink-resistance of tubes. Therefore, molecular orientation and shish-kebabs in off-axial direction were the most important reasons for significant enhancement of torsional strength in tubes during the rotation extrusion process.

An interesting conclusion found by Porter *et al.* [46] was that the mechanical property of the cylindrical materials was enhanced with the increase of helical reinforcement angle. So hoop torsional strength of PE tubes was gradually enhanced with the mandrel rotation rates increasing, as shown in figure 6. When the mandrel rotation rate was 30 r.p.m., the off-axial angle of oriented lamellae in PE and XPE tubes was 43° and 51°, and hoop torsional strength reached the maximum value (15.28 and 19.58 MPa, respectively). Therefore, the XPE-30 tubes had most excellent mechanical properties with the combined effect of rotation extrusion and Si-XLPE. With mandrel rotation, alignment of shish-kebabs induced by stable cross-linked network structures in XPE tubes along the maximum stress induced by torque force facilitated the tubes to withstand greater torsional load, and consequentially the kink-resistance of PE tubes was enhanced.

## 4. Conclusion

In our study, we have successfully attained PE tubes with excellent performance by the addition of Si-XLPE under the rotation extrusion. Meanwhile, the effects of Si-XLPE on crystalline morphology

and mechanical properties of PE tubes prepared by the micro-rotation rheometer were also revealed. During rotation extrusion, PE melts helically flowed along the axial direction, and molecular orientation and shish-kebabs in the off-axial direction were found in tubes. Although similar crystalline structures were found in PE and XPE tubes by the SEM and 2D-SAXS, orientation degree and torsional strength of XPE tubes were better than those of PE tubes. The cross-linked network structures provided by Si-XLPE in PE tubes suppressed the thermal movement and relaxation of oriented molecular chains due to intermolecular interaction; then the complicated rotation flow field promoted cross-linked network points aligning and forming more stable cluster-like shish structures during the rotation extrusion process. So shish-kebabs induced by chemical cross-linked network points of Si-XLPE were more stable than those induced by molecular chains in PE tubes. However, there was only little effect of stable shish-kebabs on significant enhancement of mechanical properties for PE tubes. Experimental results verified that molecular orientation and shish-kebabs in the off-axial direction should be primarily responsible for the remarkable enhancement of hoop torsional strength in PE tubes. With the increase of mandrel rotation rates, the off-axial angle of oriented lamellae in PE tubes increased, accompanied by enhancement of hoop torsional strength for tubes. Hoop torsional strength of XPE tubes was 19.58 MPa when the mandrel rotation rate was 30 r.p.m., while that of conventional extruded PE tubes was only 9.83 MPa, increased by 99.2%. As a consequence, PE tubes with excellent performance were prepared under the combined effect of Si-XLPE and rotation extrusion.

Data accessibility. This article does not contain any additional data.

Authors' contributions. F.S. performed the experiments and drafted the manuscript with the support from Y.L.; S.B. participated in data analysis, carried out sequence alignments, participated in the design of the study; Q.W. and S.B. conceived of the study, designed the study, coordinated the study and helped draft the manuscript. All authors revised the article and approved the version to be published and were accountable for all aspects of the work in ensuring that questions relating to the accuracy or integrity of any part of the work were appropriately investigated and resolved.

Competing interests. We have no competing interests.

Funding. This work was supported by the National Natural Science Foundation of China (nos. 51573117 and 51873131).

Acknowledgements. The authors acknowledge Dr Nie M. for providing continuous support and guidance during the research.

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
