## [Reviewer comments · Royal Society Open Science]

Review History

RSOS-182095.R0 (Original submission)

Review form: Reviewer 1

Is the manuscript scientifically sound in its present form?

Yes

Are the interpretations and conclusions justified by the results?

No

Is the language acceptable?

Yes

Is it clear how to access all supporting data?

Not Applicable

Do you have any ethical concerns with this paper?

No

Have you any concerns about statistical analyses in this paper?

Yes

Recommendation?

Accept with minor revision (please list in comments)

Comments to the Author(s)

Based on an extensive characterization study complimentary findings enable the authors to provide insight in the structure function relationship of cross linked PE particles filled HPPE tubes made via a new, elegant rotational extrusion technology. The idea is to induce non-axial orientation of self-enhanced polymer structures, improving the mechanical resilience in hoop stress.

It is evident that characterization has been done we great expertise and care, leading to figures of high quality. Also the translation of fundamental studies to product performance (hoop strength) is appreciated. However, prior to publication minor revisions have to be pursued mainly on the (over)interpretation and derived conclusions. The issues are listed below.

- Figure 1. To support and validate the interpretation on the properties of the cross-linked PEs error bars need to be included. Generally stating, from the results it can be concluded (as the authors at the end correctly do) that the most predominant improvement in properties is caused by the oriented structures so it is advised not to over-interpret the differences in crystallinity and lamellar thickness.
- Table 1. For adequate interpretation of the values of crystallinity and lamellar thickness based on DSC error bars are a MUST here as well. For the calculation of lamellar thickness via the Gibbs Thomson equation the melting temperature T_m is used. Of course the method provides an average. However, what is the melting temperature? First of all, T_m needs to be accurately described. Is it the peak temperature? Most importantly, since the sample mass varied between 6 and 9 mg, potential weight differences induce differences in the heat transfer efficiency changes, causing a thermal lag that in turn affects the peak temperature. Hence, in order to tell whether the minuscule variations of the DSC based values are statistically significant, the inclusion of error bars are a must. Next, the interpretation of the data needs to be revised. Besides the value of Δe is missing and has to be included though the discussion considers relative changes.
- Table 1. Also for the XRD based values error bars must be included, validating the interpretation. Since also here the variations in structural parameters is extremely small, applicability of the Scherrer equation in calculation of crystal size and its significance should be strengthened by the lamellar thickness derived from the long period in SAXS measurements that in fact have been performed.
- Page 15: please support the statement that the introduction of crosslinked particles suppress the thermal movement of molecular chains either experimentally (eg rheometry) or by adequate references. Thus, please place the rheometry section discussed at page 22 here. Why would the cross-linked structures preferably orient and from shishes? To support the hypothesis, including information on the crosslinking degree of the Si-XLPE seems needed. The discussion repeats on page 19, where it is stated that the chemical cross-linked structure affects the relaxation times of the matrix polymer. If the rheometry section is place forward the paper/discussion improves considerably.
- Page 17. Please support the explanation that the higher mandrel rotation rate restrains growth of oriented molecular chains and in fact promotes relaxation to the random coil. Is there break-up of the melt? Or, slip at mandrel – polymer interface?
- It is advised to run a spelling checker once more as grammar mistakes are encountered regularly.

Review form: Reviewer 2

Is the manuscript scientifically sound in its present form?

Yes

Are the interpretations and conclusions justified by the results?

Yes

Is the language acceptable?

Yes

Is it clear how to access all supporting data?

Yes

Do you have any ethical concerns with this paper?

No

Have you any concerns about statistical analyses in this paper?

No

Recommendation?

Accept with minor revision (please list in comments)

Comments to the Author(s)

In this work, the authors proposed an interesting strategy to enhance the performance of polyethylene tubes. The modified polyethylene tubes, prepared by the addition of Si-XLPE in rotation extrusion, showed improved performance. Structure-property relationship was very carefully analyzed and discussed. Comprehensive characterization and discussion were presented. The article is well-written. I only have minor comments for the authors to address.

1. Please provide the full phrase of S3M when it first appeared in the main text.
2. Based on section 2.2, the operating process can be found in a previous literature. Since this is important to this study, the authors may want to consider either briefly describe the process in the experimental section or in the supporting information.
3. Will it help to provide some detailed information about Si-XLPE material in section 2.1?
4. For equation (1) and (2), please fix the format.

Decision letter (RSOS-182095.R0)

29-Mar-2019

Dear Dr Bai:

Title: Preparation of High Performance Polyethylene Tubes under the Coexistence of Silicone Cross-linked Polyethylene and Rotation Extrusion
Manuscript ID: RSOS-182095

The editor assigned to your manuscript has now received comments from reviewers. We would

like you to revise your paper in accordance with the referee and Subject Editor suggestions which can be found below (not including confidential reports to the Editor). Please note this decision does not guarantee eventual acceptance.

Please submit your revised paper before 21-Apr-2019. Please note that the revision deadline will expire at 00.00am on this date. If we do not hear from you within this time then it will be assumed that the paper has been withdrawn. In exceptional circumstances, extensions may be possible if agreed with the Editorial Office in advance. We do not allow multiple rounds of revision so we urge you to make every effort to fully address all of the comments at this stage. If deemed necessary by the Editors, your manuscript will be sent back to one or more of the original reviewers for assessment. If the original reviewers are not available we may invite new reviewers.

Please also include the following statements alongside the other end statements. As we cannot publish your manuscript without these end statements included, if you feel that a given heading is not relevant to your paper, please nevertheless include the heading and explicitly state that it is not relevant to your work.

- Ethics statement

Please clarify whether you received ethical approval from a local ethics committee to carry out your study. If so please include details of this, including the name of the committee that gave consent in a Research Ethics section after your main text. Please also clarify whether you received informed consent for the participants to participate in the study and state this in your Research Ethics section.

OR

Please clarify whether you obtained the necessary licences and approvals from your institutional animal ethics committee before conducting your research. Please provide details of these licences and approvals in an Animal Ethics section after your main text.

OR

Please clarify whether you obtained the appropriate permissions and licences to conduct the fieldwork detailed in your study. Please provide details of these in your methods section.

- Acknowledgements

Yours sincerely,
Dr Laura Smith

Publishing Editor, Journals

On behalf of the Subject Editor Professor Anthony Stace and the Associate Editor Professor Claire Carmalt.

RSC Associate Editor:
 Comments to the Author:
 (There are no comments.)

RSC Subject Editor:
 Comments to the Author:
 (There are no comments.)

Reviewers' Comments to Author:
 Reviewer: 1

Comments to the Author(s)

Based on an extensive characterization study complimentary findings enable the authors to provide insight in the structure function relationship of cross linked PE particles filled HPPE tubes made via a new, elegant rotational extrusion technology. The idea is to induce non-axial orientation of self-enhanced polymer structures, improving the mechanical resilience in hoop stress.

It is evident that characterization has been done we great expertise and care, leading to figures of high quality. Also the translation of fundamental studies to product performance (hoop strength) is appreciated. However, prior to publication minor revisions have to be pursued mainly on the (over)interpretation and derived conclusions. The issues are listed below.

- Figure 1. To support and validate the interpretation on the properties of the cross-linked PEs error bars need to be included. Generally stating, from the results it can be concluded (as the authors at the end correctly do) that the most predominant improvement in properties is caused by the oriented structures so it is advised not to over-interpret the differences in crystallinity and lamellar thickness.
- Table 1. For adequate interpretation of the values of crystallinity and lamellar thickness based on DSC error bars are a MUST here as well. For the calculation of lamellar thickness via the Gibbs Thomson equation the melting temperature T_m is used. Of course the method provides an average. However, what is the melting temperature? First of all, T_m needs to be accurately described. Is it the peak temperature? Most importantly, since the sample mass varied between 6 and 9 mg, potential weight differences induce differences in the heat transfer efficiency changes, causing a thermal lag that in turn affects the peak temperature. Hence, in order to tell whether the minuscule variations of the DSC based values are statistically significant, the inclusion of error bars are a must. Next, the interpretation of the data needs to be revised. Besides the value of Δe is missing and has to be included though the discussion considers relative changes.
- Table 1. Also for the XRD based values error bars must be included, validating the interpretation. Since also here the variations in structural parameters is extremely small,

applicability of the Scherrer equation in calculation of crystal size and its significance should be strengthened by the lamellar thickness derived from the long period in SAXS measurements that in fact have been performed.

- Page 15: please support the statement that the introduction of crosslinked particles suppress the thermal movement of molecular chains either experimentally (eg rheometry) or by adequate references. Thus, please place the rheometry section discussed at page 22 here. Why would the cross-linked structures preferably orient and from shishes? To support the hypothesis, including information on the crosslinking degree of the Si-XLPE seems needed. The discussion repeats on page 19, where it is stated that the chemical cross-linked structure affects the relaxation times of the matrix polymer. If the rheometry section is place forward the paper/discussion improves considerably.
- Page 17. Please support the explanation that the higher mandrel rotation rate restrains growth of oriented molecular chains and in fact promotes relaxation to the random coil. Is there break-up of the melt? Or, slip at mandrel – polymer interface?
- It is advised to run a spelling checker once more as grammar mistakes are encountered regularly.

Reviewer: 2

Comments to the Author(s)

In this work, the authors proposed an interesting strategy to enhance the performance of polyethylene tubes. The modified polyethylene tubes, prepared by the addition of Si-XLPE in rotation extrusion, showed improved performance. Structure-property relationship was very carefully analyzed and discussed. Comprehensive characterization and discussion were presented. The article is well-written. I only have minor comments for the authors to address.

1. Please provide the full phrase of S3M when it first appeared in the main text.
2. Based on section 2.2, the operating process can be found in a previous literature. Since this is important to this study, the authors may want to consider either briefly describe the process in the experimental section or in the supporting information.
3. Will it help to provide some detailed information about Si-XLPE material in section 2.1?
4. For equation (1) and (2), please fix the format.

Author's Response to Decision Letter for (RSOS-182095.R0)

See Appendix A.

Decision letter (RSOS-182095.R1)

15-Apr-2019

Dear Dr Bai:

Title: Preparation of High Performance Polyethylene Tubes under the Coexistence of Silicone Cross-linked Polyethylene and Rotation Extrusion
Manuscript ID: RSOS-182095.R1

It is a pleasure to accept your manuscript in its current form for publication in Royal Society Open Science. The chemistry content of Royal Society Open Science is published in collaboration with the Royal Society of Chemistry.

On behalf of the Subject Editor Professor Anthony Stace and the Associate Editor Professor Claire Carmalt.

Appendix A

THE STATE KEY LABORATORY OF POLYMER MATERIALS ENGINEERING
SICHUAN UNIVERSITY

Chengdu, Sichuan 610065, China Tel and Fax: +86-28-85405133

Dear Editors and Reviewers:

Thank you for your e-mail and the attached comments on our manuscript entitled “*Preparation of High Performance Polyethylene Tubes under the Coexistence of Silicone Cross-linked Polyethylene and Rotation Extrusion*” (RSOS-182095). Those comments and suggestions are really helpful to improve our manuscript quality. All the changes we made were clearly marked in red color in the revised manuscript and the point-by-point responses to the reviewer comments are included hereinafter.

Response to Reviewer 1:

Q1. Figure 1. To support and validate the interpretation on the properties of the cross-linked PEs error bars need to be included. Generally stating, from the results it can be concluded (as the authors at the end correctly do) that the most predominant improvement in properties is caused by the oriented structures so it is advised not to over-interpret the differences in crystallinity and lamellar thickness.

A: Thank you for your instructive suggestions. The properties of the XLPE error bars have been added into figure 1 in our revised manuscript. Indeed, the crystallinity and lamellar thickness displayed no obvious impact on the reinforcement of the tubes and thus we have further shortened the corresponding discussions in the revised manuscript to emphasize the importance of the oriented structures.

Q2. Table 1. For adequate interpretation of the values of crystallinity and lamellar thickness based on DSC error bars are a MUST here as well. For the calculation of

lamellar thickness via the Gibbs Thomson equation the melting temperature T_m is used. Of course the method provides an average. However, what is the melting temperature? First of all, T_m needs to be accurately described. Is it the peak temperature? Most importantly, since the sample mass varied between 6 and 9 mg, potential weight differences induce differences in the heat transfer efficiency changes, causing a thermal lag that in turn affects the peak temperature. Hence, in order to tell whether the minuscule variations of the DSC based values are statistically significant, the inclusion of error bars are a must. Next, the interpretation of the data needs to be revised. Besides the value of σ_e is missing and has to be included though the discussion considers relative changes.

A: You are right that the variation in mass can cause a thermal lag and thus lead to changes of the peak temperature. In our study, the melting temperatures are all defined as peak temperature in the melting curve and hence we have added the error bars for T_m and the corresponding L_c in the Table 1. Finally, the σ_e has also been added in the revised manuscript. Thank you again for your suggestion.

Q3. *Table 1. Also for the XRD based values error bars must be included, validating the interpretation. Since also here the variations in structural parameters is extremely small, applicability of the Scherrer equation in calculation of crystal size and its significance should be strengthened by the lamellar thickness derived from the long period in SAXS measurements that in fact have been performed.*

A: As suggested, we have added the error bars for the values of crystallinity and

lamellar thickness based on XRD and the long period of SAXS, respectively. Despite crystalline data obtained from DSC and XRD slightly differed from each other, both data displayed the same trend that with increasing mandrel rotation rate, the crystallinity and lamellar thickness firstly increased and then slightly dropped once the rotation rate exceeded 30 rpm. These results fit well with the explanation that high-speed rotation can generate large amount of shear heat to facilitate the relaxation of the stretched molecules to the random coil. The corresponding discussions have been added in the revised manuscript.

Q4. Page 15: please support the statement that the introduction of crosslinked particles suppress the thermal movement of molecular chains either experimentally (eg rheometry) or by adequate references. Thus, please place the rheometry section discussed at page 22 here. Why would the cross-linked structures preferably orient and from shishes? To support the hypothesis, including information on the crosslinking degree of the Si-XLPE seems needed. The discussion repeats on page 19, where it is stated that the chemical cross-linked structure affects the relaxation times of the matrix polymer. If the rheometry section is place forward the paper/discussion improves considerably.

A: Thank you for your comments. As demonstrated by many open literature (1. Liu D, Tian N, Cui K, Zhou W, Li X, Li L. 2013. *Macromolecules*, 46, 3435-3443. (DOI: 10.1021/ma400024m) 2. Tian Y, Zhu C, Gong J, Yang S, Ma J, Xu J. 2014. *Polymer*, 55(16), 4299-4306. (<https://doi.org/10.1016/j.polymer.2014.06.056>)), the formation of

shish is a competing process between stretch and relaxation of the molecular chains. The mobility of the molecular chains is the key factor that determines the final configuration. In rheology, the mobility of the molecular chains can be reflected by the value of relaxation time λ . According to the frequency sweep at 210 °C, the relaxation time of the PE and XPE specimens is 0.47s and 1.04s, respectively. Apparently, the molecular chains of the XPE are harder to move and accordingly are capable to maintain their stretched configuration under the subsequent cooling procedure. Correspondingly, the related discussions as well as the information on the crosslinking degree of the Si-XLPE have also been added in the revised manuscript. Thank you for your suggestion.

Q5. Page 17. Please support the explanation that the higher mandrel rotation rate restrains growth of oriented molecular chains and in fact promotes relaxation to the random coil. Is there break-up of the melt? Or, slip at mandrel – polymer interface?

A: Thank you for your instructive suggestions. As discussed earlier, the formation of shish involves two competing procedure, i.e. the stretch and relaxation of the molecular chains. The former one is the result of stress filed which is thermodynamic instability and the latter one is derived from the molecular thermal motion. With increasing rotation rate, a large amount of heat was generated inside the polyethylene due to the shear between mandrel and melt. Since polymer is a poor conductor of heat, shear heat was keen to concentrate in local area, and thus accelerated the thermal motion of molecular chains where the stretching molecular chains relaxed into the

random coil. (1. Cho S, Jeong S, Kim JM, Baig C. 2017. SCI REP-UK 7, 9004.

(<https://doi.org/10.1038/s41598-017-08712-5>) 2. Ramos J, Vega J, Martínez-Salazar

J. 2017 Eup. Polym. J. 99, 298-331. (<https://doi.org/10.1016/j.eurpolymj.2017.12.027>)

As far as we know, the shear rate is not too high to cause obvious break-up of the melt or slip at interface, because despite the fast speed, the outer diameter of the mandrel is only 4mm. The explanation has been included in the revised manuscript.

Q6. It is advised to run a spelling checker once more as grammar mistakes are encountered regularly.

A: Thank you for your comment, and we have carefully revised our manuscript and polished the English writing in our revised manuscript. The inappropriate expression and mistakes have been corrected.

Response to Reviewer 2:

Q1. Please provide the full phrase of S³M when it first appeared in the main text.

A: We have added the full phrase of S³M in our revised manuscript. The full phase of S³M is solid-state shear mechanochemical.

Q2. Based on section 2.2, the operating process can be found in a previous literature.

Since this is important to this study, the authors may want to consider either briefly describe the process in the experimental section or in the supporting information.

A: Indeed, the operating process has great impact on the final properties of the XLPE.

In order to achieve comprehensive understanding of our work for the potential readers, it is necessary to provide as an intuitional description in the experimental section of the revised manuscript. Thank you for your suggestion.

Q3. Will it help to provide some detailed information about Si-XLPE material in section 2.1?

A: Thank you for your instructive suggestions. The detailed information about Si-XLPE material such as gel content and melt flow rate has been added in section 2.1 of the revised manuscript.

Q4. For equation (1) and (2), please fix the format.

A: We have carefully fixed the format of equation (1) and (2) in our revised manuscript.

According to the comments and suggestions of the reviewers, we have accordingly made revisions on our manuscript marked in red color, as shown in our revised manuscript. The revised manuscript has been submitted online for your further consideration. We are looking forward to hearing from you. Thank you for your cooperation.

Kind regards,

Sincerely yours,

Shibing Bai

THE STATE KEY LABORATORY OF POLYMER MATERIALS ENGINEERING
SICHUAN UNIVERSITY

Chengdu, Sichuan 610065, China Tel and Fax: +86-28-85405133

State Key Laboratory of Polymer Materials Engineering

Polymer Research Institute, Sichuan University

Chengdu 610065, China

Tel: 86-28-85405136

Fax: 86-28-85402465

E-mail: baishibing@scu.edu.cn